# Clinical and Virological Features of Patients Hospitalized with Different Types of COVID-19 Vaccination in Mexico City

**DOI:** 10.3390/vaccines10081181

**Published:** 2022-07-26

**Authors:** Alejandra Hernández-Terán, Pamela Garcíadiego-Fossas, Marco Villanueva-Reza, Celia Boukadida, Blanca Taboada, Eduardo Porras, Victor Ahumada-Topete, Kathia Elizabeth Tapia-Diaz, Margarita Matías-Florentino, Marissa Pérez-García, Santiago Ávila-Ríos, Fidencio Mejía-Nepomuceno, Ricardo Serna-Muñoz, Fortunato Juárez-Hernández, María Eugenia Jiménez-Corona, Eduardo Becerril-Vargas, Omar Barreto, Jose Arturo Martínez-Orozco, Rogelio Pérez-Padilla, Carlos F. Arias, Joel Armando Vázquez-Pérez

**Affiliations:** 1Departamento de Investigación en Tabaquismo y EPOC, Instituto Nacional de Enfermedades Respiratorias Ismael Cosío Villegas, INER, Mexico City 14080, Mexico; alejandra.hdezteran@comunidad.unam.mx (A.H.-T.); biolfimene@gmail.com (F.M.-N.); serna.971@gmail.com (R.S.-M.); perezpad@gmail.com (R.P.-P.); 2Departamento de Unidad de Epidemiología Hospitalaria e Infectología, Instituto Nacional de Enfermedades Respiratorias Ismael Cosío Villegas, INER, Mexico City 14080, Mexico; infectologia13@gmail.com (P.G.-F.); marco.vilre@gmail.com (M.V.-R.); edxa86@gmail.com (E.P.); victor.ahumada@cieni.org.mx (V.A.-T.); 3Centro de Investigación en Enfermedades Infecciosas, CIENI, Instituto Nacional de Enfermedades Respiratorias Ismael Cosío Villegas, INER, Mexico City 14080, Mexico; celia.boukadida@cieni.org.mx (C.B.); kathia.tapia.cieni@gmail.com (K.E.T.-D.); margarita.matias@cieni.org.mx (M.M.-F.); marissa.perez@cieni.org.mx (M.P.-G.); santiago.avila@cieni.org.mx (S.Á.-R.); 4Departamento de Genética del Desarrollo y Fisiología Molecular, Instituto de Biotecnología, Universidad Nacional Autónoma de México, Cuernavaca 62210, Mexico; blancai.taboada@gmail.com; 5Departamento de Imagenología, Instituto Nacional de Enfermedades Respiratorias Ismael Cosío Villegas, INER, Mexico City 14080, Mexico; drjuarez.radiologo@gmail.com; 6Departamento de Epidemiología, Instituto Nacional de Cardiología Ignacio Chávez, Mexico City 14080, Mexico; mejimenez777@gmail.com; 7Laboratorio de Microbiología, Instituto Nacional de Enfermedades Respiratorias Ismael Cosío Villegas, INER, Mexico City 14080, Mexico; edobec.var@gmail.com (E.B.-V.); drjamoinfectologia@gmail.com (J.A.M.-O.); 8Coordinación de Atención Médica, Instituto Nacional de Enfermedades Respiratorias Ismael Cosío Villegas, INER, Mexico City 14080, Mexico; omar_barreto10@hotmail.com

**Keywords:** COVID-19, vaccination schemes, SARS-CoV-2 lineages, COVID-19 severity, comorbidities in COVID-19

## Abstract

Coronavirus disease 2019 (COVID-19) vaccines effectively protect against severe disease and death. However, the impact of the vaccine used, viral variants, and host factors on disease severity remain poorly understood. This work aimed to compare COVID-19 clinical presentations and outcomes in vaccinated and unvaccinated patients in Mexico City. From March to September 2021, clinical, demographic characteristics, and viral variants were obtained from 1014 individuals with a documented SARS-CoV-2 infection. We compared unvaccinated, partially vaccinated, and fully vaccinated patients, stratifying by age groups. We also fitted multivariate statistical models to evaluate the impact of vaccination status, SARS-CoV-2 lineages, vaccine types, and clinical parameters. Most hospitalized patients were unvaccinated. In patients over 61 years old, mortality was significantly higher in unvaccinated compared to fully vaccinated individuals. In patients aged 31 to 60 years, vaccinated patients were more likely to be outpatients (46%) than unvaccinated individuals (6.1%). We found immune disease and age above 61 years old to be risk factors, while full vaccination was found to be the most protective factor against in-hospital death. This study suggests that vaccination is essential to reduce mortality in a comorbid population such as that of Mexico.

## 1. Introduction

Severe acute respiratory syndrome coronavirus 2 (SARS-CoV-2) was first detected in Wuhan, China, in December 2019 [1]; since then, more than 500 million people have been infected, and over 6 million have died worldwide [2]. SARS-CoV-2 has evolved, and several mutations throughout the genome have been detected worldwide [3]. Genomic mutations are anticipated events during virus replication, and although most mutations are expected to be neutral, some can confer a fitness advantage and be fixed in the viral genome [4,5]. The accumulation of mutations over time generates new SARS-CoV-2 variants [6]. The World Health Organization (WHO) has classified them as variants of concern (VOC), variants of interest (VOI), and variants under monitoring (VUM) (WHO, tracking SARS-CoV-2 variants). VOC are SARS-CoV-2 variants with increased transmissibility, virulence, or those that resist the effectiveness of vaccines, therapeutics, diagnostics, or public health and social measures [6].

In Mexico, several variants classified as VOC, VOI, or VUM have been detected, particularly Delta, Alpha, Gamma, Mu [7], and, more recently, Omicron [8]. Moreover, B.1.1.519 was detected as the predominant lineage in Mexico during late 2020 and the first months of 2021 [9]. The B.1.1.519 lineage was dominant during the second COVID-19 wave in Mexico and may have been associated with an increased risk of severe and fatal outcomes [10].

Vaccination against COVID-19 began in December 2020 with health personnel and people over 60 years old, followed by people over 50. By the end of September 2021, 101,190,484 doses of vaccines were applied, and almost 50% of the population over 18 years old had received at least one dose of the vaccine [11]. Importantly, COVID-19 vaccines effectively prevent severe disease and death even in patients with comorbidities [12]. Previous reports have highlighted the role of immune diseases as a risk factor in the progression to severe COVID-19 and fatal outcomes [13]. Since this population is at risk, vaccination is a safe and effective option to avoid progression to severe disease and to avoid exacerbations or complications from underlying conditions.

Patients with autoimmune conditions could be one of the populations that benefits the most from vaccination. It has been reported that immunocompromised patients with conditions such as hematological or solid cancer, immune-mediated inflammatory disorders, and even organ transplant receptors show adequate seroconversion after a second dose of the COVID-19 vaccine [14]. Regarding the risks of vaccination in patients with immune disorders, recently, a small cohort of patients with myasthenia gravis did not find any worsening symptoms (90.9%) after vaccination. [15]

In accordance with this, fully vaccinated individuals are less likely to experience severe disease or death than unvaccinated persons with the same medical conditions [16]. Nonetheless, vaccines are less effective at protecting individuals from SARS-CoV-2 infections, and this partial protection seems to wane after a few months, leading to infections in vaccinated individuals (breakthrough infections) [17]. Furthermore, one of the most critical factors behind the changing vaccine effectiveness is the viral variability and the mutations that affect the recognition of the neutralizing antibodies elicited by vaccination or previous infections [18,19]. Understanding the impact of vaccination, viral variants, and host factors on disease severity is critical to guiding COVID-19 vaccination campaigns and protective measures. Here, we analyzed the main risk factors for severe COVID-19, comparing the outcomes and clinical presentation of vaccinated and unvaccinated patients and analyzing the effects of both SARS-CoV-2 lineages and vaccine types in a tertiary hospital in Mexico City from March to September 2021.

## 2. Material and Methods

### 2.1. Study Participants and Clinical Data

We analyzed 1014 patients with SARS-CoV-2 infection who received medical attention from March to September 2021 at the Instituto Nacional de Enfermedades Respiratorias (INER). For all patients, demographic data, clinical symptoms, and laboratory and outcome-related information were obtained from electronic medical records. Clinical management was performed according to the standards of care and attending physicians. This study was reviewed and approved by the Science, Biosecurity, and Bioethics Committee of the Instituto Nacional de Enfermedades Respiratorias (B-10-20) and informed consent for the recovery, storage, and use of biological remnants for research purposes was requested.

### 2.2. SARS-CoV-2 Diagnostics

Oropharyngeal and/or nasopharyngeal swabs were collected, and the diagnosis was made using validated RT-qPCR protocols for SARS-CoV-2 RNA detection, approved by the World Health Organization (WHO).

### 2.3. RNA Extraction and Sequencing

#### 2.3.1. Complete Genome Sequencing

Viral nucleic acid extraction was performed using a MagNa Pure L.C. 2-0 system (Roche, Indianapolis, OH, USA) or QIAamp viral RNA Minikit (Qiagen, Hilden, Germany). Libraries for the whole-genome sequencing of SARS-CoV-2 were generated using the protocol developed by the ARTIC Network (https://artic.network/2-protocols.html, accessed on 10 September 2020) or a long-amplicon-based method [20]. Libraries were sequenced on a MiSeq sequencing platform using a 2 × 150-cycle or a NextSeq 500 platform using 2 × 150-cycle mid-output kits to obtain paired-end reads (Illumina, San Diego, CA, USA). The DRAGEN COVIDSeq Test Pipeline on BaseSpace Sequence Hub performed the analysis, mapping, and consensus.

#### 2.3.2. Spike Partial Sequencing

Partial sequencing of the Spike segment was performed by Sanger sequencing in samples with incomplete genome or samples with cycle threshold (Ct) values above 28. Briefly, 956 bp amplicons (944–1900 nucleotide sequence, 315–633 aa) were obtained using specific primers for SARS-CoV-2:

SF3 CTTCTAACTTTAGAGTCCAACC and SR4 GCCAAGTAGGAGTAAGTTGAT.

The amplicons were sequenced in both directions with the same primers. Sequencing reactions were performed with a BigDye Terminator v3.1 (Life Technologies, Carlsbad, CA, USA) as instructed by the manufacturer. Sequences were obtained by capillary electrophoresis using an ABI Prism 3500 Genetic Analyzer (Life Technologies) and were assembled using MEGA 10.0 [21]. The sequences mentioned above can be found at GenBank (accession numbers: ON158371–ON158443).

### 2.4. Phylogenetic Analysis

To perform phylogenetic analysis, we analyzed complete Spike sequences (*n* = 311) and, additionally, we selected 230 sequences available on the GISAID platform. Sequence alignments were created with MAFFT V7 [22] and edited with MEGA 10.0 [21]. A maximum-likelihood tree was constructed for the whole Spike sequence using MEGA 10.0. The General Time-Reversible (GTR) model was selected with five-parameter gamma-distributed rates and 1000 bootstrap replicates. Edition of the trees was made using FigTree [23].

### 2.5. Statistical Analyses

To evaluate the impact on clinical outcomes, we divided our cohort into three groups: (1) unvaccinated, (2) partially vaccinated, and (3) fully vaccinated (for more information, see Appendix A). As advanced age is a significant driver of severe COVID-19 [24], we categorized patients into three groups: <30 years old, 31–60 years old, and >61 years old. In addition, the severity of the disease was categorized according to the NIH COVID-19 treatment guidelines [25] into four groups: (1) mild illness, (2) moderate illness, (3) severe illness, and (4) critical illness.

All statistical analyses were performed using R v4.0.2 in RStudio v1.3.1 [26] and the packages ggplot2 (v3.3.3) [27], stats (v3.6.2) [26], survival (v3.2.13) [28], and survminer (v0.4.9) [29]. We used stacked bar plots, density plots, pie charts, or heatmaps constructed in the ggplot2 package to represent the data proportion. For numerical data representation, we used boxplots constructed in the ggplot2 package. For all cases, categorical variables were statistically compared using Chi-square or Fisher’s exact test and continuous variables using Wilcoxon rank-sum test in R base.

To predict the clinical outcome (deceased vs. recovered), we constructed generalized linear models (GLM) using a binomial family with a logit link in R base. We fitted one model as a function of vaccination status and another model as a function of the clinical parameters. All models were adjusted by comorbidities, age, and sex. Odds ratios and 95% CI for all selected variables were calculated and plotted in the final model. Finally, to detect variables that significantly affect survival probability, we calculated Kaplan–Meier curves coupled with the Cox proportional hazard models to determine the factors affecting survival, using the survival package and plotted with the survminer package. We used the hospitalization length in days as a time variable for all curves, the outcome (deceased or recovered) standardized at 28 days as a dependent variable, and the tested variables as exposure. Finally, for both the GLM and the Kaplan–Meier curves, we categorized the numerical variables (e.g., age, D-dimer) into three groups according to the data distribution: (1) all values under the 25th percentile, (2) values between the 25th and 75th percentiles, and (3) values above the 75th percentile.

## 3. Results

### 3.1. Clinical and Demographic Characteristics of the Cohort

A total of 1014 patients admitted to INER from March to September 2021 positive for SARS-CoV-2 infection were included. In total, 124 (12%) were outpatients, and 890 (88%) were hospitalized due to respiratory failure. Moreover, 48 patients in the cohort had incomplete information; thus, only 111 out of 124 outpatients and 855 out of 890 hospitalized patients were included in the demographic analysis (Table 1).

The median age was 39.5 years for outpatients, and 64% of the patients were either partially (28.8%) or fully (36%) vaccinated. Pfizer, Sinovac, and AstraZeneca were the most commonly administered vaccines for all outpatients. In addition, 13.5% of the patients had at least one comorbidity, with obesity being the most common (14.4%), particularly in unvaccinated (12.8%) and partially vaccinated (21.8%) patients. For fully vaccinated outpatients, the most common comorbidity was hypertension (12.5%). For hospitalized patients (Table 1), the median age was 52. Most patients were unvaccinated (66.9%), while 20.8% were partially vaccinated and 12.2% were fully vaccinated. Sinovac, AstraZeneca, Pfizer, and Cansino were the most commonly administered vaccines for hospitalized patients, and 34.6% had at least one comorbidity. In general, fully vaccinated patients were significantly older and showed a higher prevalence of diabetes, hypertension, and three or more comorbidities.

### 3.2. Impact of Vaccination Status and Age on the Severity and Clinical Parameters of COVID-19 Patients

Due to the timing of the vaccination campaign in Mexico, most of the vaccinated patients were received from June to August, while unvaccinated patients arrived at the hospital in equal proportions during the sampling months (Appendix A).

In general, we found that a large proportion of patients were classified either as severe or critical, regardless of age and vaccination status (Figure 1A). Nonetheless, the highest proportion of patients with mild disease was found in fully vaccinated patients under 30 years old (Appendix A), while the highest proportion of critical patients was unvaccinated older than 61 years old (Appendix A). It is essential to highlight that for older patients (>61 years), the vaccination status seems to be associated with less disease severity, reducing the number of critical patients (Appendix A, unvaccinated against vaccinated patients) and slightly increasing the number of patients with mild disease in the vaccinated groups (Appendix A). Nonetheless, we detected that, although most of the patients were subjected to mechanical ventilation, the highest proportion of intubated patients was in the unvaccinated group (Figure 1C). Finally, we found that, regardless of the vaccination status, patients older than 61 years exhibited the highest values for blood urea nitrogen (BUN) (Figure 2), urea, creatinine (only statistically significant in unvaccinated patients), and D-dimer. In addition, patients in the age groups of <30 and 31–60 showed the highest values for lymphocytes, hematocrit, and albumin (only statistically significant in unvaccinated patients).

### 3.3. Impact of Different Vaccines on Severity of Disease

To eliminate the age factor (Appendix A), we kept only fully vaccinated patients above 61 years old (N = 88) and tested for differences in the severity of the disease, O_2_ requirement, and severity indexes (Figure 3C–E). For instance, we found the highest proportion of patients with mild disease in the Pfizer group (Figure 3C and Appendix A). Furthermore, patients vaccinated with AstraZeneca, Sinovac, and Sputnik had mostly critical disease. Moreover, AstraZeneca was the group with the highest proportion of patients with mechanical ventilation (66%), while Pfizer was the group with the highest proportion of nasal cannula (16.6%). Finally, we found that the Sinovac group had significantly higher values for SOFA (Figure 3E) compared to Pfizer and AstraZeneca.

### 3.4. Clinical Factors Affecting COVID-19 Clinical Outcome

Of the hospitalized patients, 77.1% were discharged fully recovered, while 21.2% died due to complications (Table 2), with the highest proportion of deceased patients in the unvaccinated group (70.8%). Fully vaccinated patients also had the highest median of APACHE II. Moreover, we found that the presence of immune disease (OR: 3.12, 95% CI: 1.09–8.34, *p* = 0.02), age above 61 years old (OR: 3.51, 95% CI: 2.3–5.2, *p* = 5.9 × 10^−10^), platelet count under 175 × 10^9^/l (OR: 1.49, 95% CI: 0.98–2.2, *p* = 0.05), lactate dehydrogenase (LDH) >600 ug/L (OR: 1.89, 95% CI: 1.2–2.7, *p* = 0.001), and D-dimer >1.8 ug/mL (OR: 1.45, 95% CI: 0.98–2.1, *p* = 0.05) were the factors associated with a higher probability of death, while being fully vaccinated was associated with improved outcomes (OR: 0.25, 95% CI: 0.12–0.46, *p* = 2.89 × 10^−5^) (Figure 4 and Appendix A). Finally, age and D-dimer above 1.87 µg/mL were also found to correlate with mortality in the Kaplan–Meier curves (Figure 3C).

### 3.5. SARS-CoV-2 Lineages

We identified the lineages of 386 (38%) samples by partial or complete genome sequencing. From this subset of samples, we detected VOC such as Alpha (2.3%), Gamma (7.2%), Delta (68.6%), and Mu (3.6%), originally designated as VOI, as well as sequences belonging to the lineage B.1.1.519 (14.7%). Notably, the SARS-CoV-2 variant distribution displayed temporal variations, with B.1.1.519 being more prevalent in March and April 2021 and Delta becoming the predominant lineage in July nationwide, as well as in the State of Mexico, and Mexico City (Figure 5A,C and Appendix A). It is worth mentioning that the number of critical and severe patients (green and purple colors) seems to decrease as the vaccination increases (Figure 5D). Although we found no significant association between disease severity and lineages (Appendix A), we noted that diarrhea tended to be associated more frequently with B.1.1.519 infections (Figure 5B). Finally, as expected, the phylogenetic analyses showed specific clustering by lineages, with the Delta variant representing 69% of the samples (Appendix A). The sequences of patients with different vaccination statuses or severity classifications did not form specific clusters, suggesting no association between SARS-CoV-2 lineages and vaccination status or COVID-19 severity.

## 4. Discussion

Vaccines have significantly improved the COVID-19 survival rate around the world, but they are less effective at protecting against SARS-CoV-2 infections, leading to primarily mild infections in vaccinated individuals. Understanding the impact of viral variants and host factors on disease severity is critical to guide COVID-19 vaccination campaigns and the implementation of protective measures. In this retrospective study, we analyzed the severity of COVID-19 infections in a group of individuals who had been vaccinated with all different types of vaccines applied to the Mexican population. We compared this group with unvaccinated individuals and performed additional analyses, including virological features obtained from genomic surveillance.

It is known that host factors such as advanced age and the presence of comorbidities significantly worsen clinical outcomes for COVID-19 patients [30]. The Mexican population is characterized by a high prevalence of comorbidities [31], which is reflected in our cohort, with diabetes (20.5%), hypertension (25.5%), and obesity (39.4%) being the most common. Moreover, it is important to highlight that as a result of the Mexican vaccination campaign, focused on older individuals, the prevalence of comorbidities in our cohort was higher in fully vaccinated patients (Table 1), with more than 90% presenting at least one comorbidity and over 75% being older than 60 years of age. Some studies have shown that comorbidities and advanced age can also affect vaccines’ effectiveness [12]. Moreover, in both COVID-19 [6,24,32] and influenza disease [33], it has been shown that advanced age is related to a lower immune response to vaccination. Furthermore, a recent study showed that vaccinated patients with comorbidities and older age are more likely to develop severe disease [6]. This is similar to our study, where we observed that fully vaccinated patients presented the highest values of severity on the APACHE II scale (Table 2). Nonetheless, compared to unvaccinated individuals, the vaccinated group showed the lowest mortality rate (Table 2) and required mechanical ventilation less frequently (Figure 1C). These observations confirm that although the vaccines’ protective effect may be reduced in individuals with risk factors, they still provide protection against fatal outcomes.

Comorbidities such as diabetes and hypertension are risk factors that have been demonstrated to be associated with higher mortality [34]. This could be associated with the fact that depending on the comorbidity, the damage can be due to chronic end-organ injury or acute decompensation [34]. Comorbidities should be individualized, and in several studies, the odds ratio for mortality is different according to the comorbidity [35].

The impact of vaccination can be observed in lowering mortality despite having comorbidities [35]. One example is neuromuscular disorders such as myasthenia gravis, which has a high mortality percentage (44%) in patients with COVID-19—a percentage that was lowered with vaccination with a low risk of adverse events [36,37].

Moreover, we found that a significant proportion of patients had either severe or critical outcomes, regardless of age and vaccination status (Figure 1A). Although this could be a consequence of studying a cohort of patients in a tertiary hospital limited to the attention of critically ill individuals, it also could be the result of the presence of comorbidities and host genetic factors [38,39]. However, it is essential to highlight that the highest proportion of patients with mild disease was in those fully vaccinated and under 30 years of age (Appendix A), evidencing the combined protective effect of younger ages and complete vaccination schemes [40].

In addition, we found that regardless of the vaccination status, patients older than 61 years exhibited the highest values for markers of disease severity (e.g., BUN, urea, creatinine, and D-dimer), which are commonly found elevated in individuals with acute infections [41]. Moreover, urea, BUN, and creatinine are also markers for renal damage and acute kidney injury (AKI), which are conditions that strongly impact mortality [42]. Additionally, hypoalbuminemia, found in older COVID-19 patients, could be a manifestation of a pro-inflammatory state [43]. Finally, high levels of D-dimer have been found to be related to mortality in patients with infection or sepsis [44].

It is essential to point out that this study was not designed to assess vaccine effectiveness against COVID-19. However, Mexico is one of the few countries that used seven different vaccines, offering a unique opportunity to compare these vaccine types and to look for differences in severity and/or mortality. In this study, by analyzing patients solely above 61 years old, we found a high proportion of critical patients, but low mortality rates, regardless of the vaccine applied (Appendix A). Nonetheless, our findings of a lower SOFA index, higher proportion of patients with mild disease, and higher usage of nasal cannula in patients vaccinated with the Pfizer vaccine may suggest a lower severity of disease in the group that received this vaccine. Some studies have shown that the vaccination efficacy of mRNA vaccines does not decrease in individuals with comorbidities [45]. Still, no causal associations can be made from this finding, and more investigations are needed to confirm whether there is a difference in severity associated with the vaccines and what the factors behind it are.

Understanding the impact of SARS-CoV-2 lineages on disease severity in both unvaccinated and vaccinated individuals is of major importance, especially since such variants could still emerge due to the evolution pressure driven by population immunity [5,46,47]. We detected a tendency in patients > 60 years old infected with either Delta or B.1.1.519 lineages to course a more severe disease (Appendix A). Although this finding was not statistically significant in our study, other studies have reported more severe disease in infections associated with these lineages [10,48]. In particular, mutations in the spike (S) protein of the virus that confer a higher affinity of the S1 domain to the angiotensin converter enzyme 2 (ACE2) are one of the main reasons behind the higher pathogenicity of the Delta variant [48].

Regarding symptomatology, there were no remarkable differences except for infections with the B.1.1.519 lineage, since patients reported diarrhea more frequently (Figure 5B). This contrasts with other studies that found that fever, dyspnea, and sore throat were more prevalent in patients infected with Delta [6,49]. However, a previous study found that dyspnea, cyanosis, and particularly diarrhea were the most frequent symptoms in patients infected with the B.1.1.519 lineage in Mexico [10].

Moreover, previous studies analyzing large datasets have found differences in disease severity among patients infected with different lineages [50,51]. Nonetheless, studies with small datasets have failed in detecting such differences [52,53]. Altogether, the reduced sample size and the temporal variation of the lineages may have affected our results and hinder comparative analyses between lineages and vaccination status or disease severity.

We determined the main factors that lead to fatal outcomes in our cohort. According to the fitted generalized linear models (GLMs) (Figure 4), full vaccination was found to be a protective factor against death by COVID-19. On the contrary, platelets < 175 × 10^9^/L, age above 61 years old, immune disease, D-dimer > 1.8 µg/mL, and lactate dehydrogenase (LDH) above 600 µg/L were found to be risk factors. Notably, thrombocytopenia and high levels of LDH have been found to be associated with severe COVID-19 in other studies [54,55].

Furthermore, contrary to other studies [56], we identified advanced age as a risk factor in COVID-19 patients. It is known that the decline of physiologic functions and the immunosenescence associated with increasing age can negatively affect the host’s response to infectious diseases [6,24,32]. We hypothesize, then, that complete vaccination may diminish age-negative effects, since we found that despite most of the fully vaccinated patients being >61 years old (78%), 83.3% were discharged from the hospital.

## 5. Conclusions

Altogether, comorbidities and advanced age were the main risk factors, and complete vaccination schemes with different vaccine types were the most significant protective factors against death by COVID-19. Moreover, we did not find strong associations between a particular lineage (Alpha, Gamma, B.1.1.519, and Delta) and disease severity, highlighting the predominant role of host factors such as age, comorbidities, and vaccination status on COVID-19 outcomes.

Complete vaccination schemes for the whole population are crucial to reducing hospitalization and death. Nevertheless, public health policy should also focus on the control of comorbidities in the long term, which will improve clinical outcomes not only for COVID-19, but also for other current and emerging diseases. Genomic surveillance is essential to identify circulating VOC, which can help to guide the implementation of targeted epidemiological control measures and laboratory characterization for diagnostic and research purposes.

## Figures and Tables

**Figure 1 vaccines-10-01181-f001:**
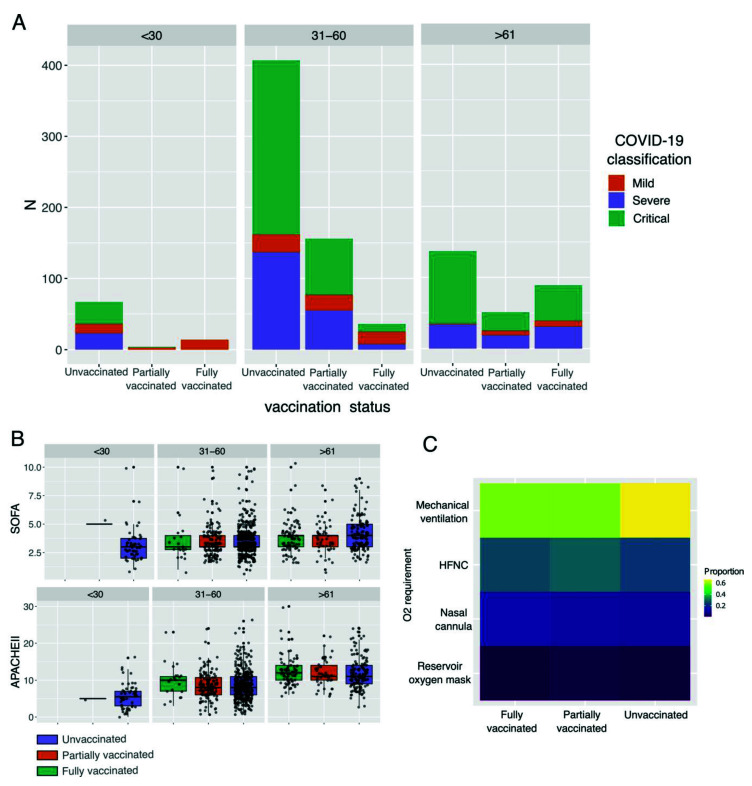
Impact of vaccination status and age in severity and clinical parameters in COVID-19 patients. (**A**) Bar plot shows the proportion of COVID-19 severity classification among unvaccinated, partially vaccinated, and fully vaccinated patients. Statistically significant differences are available in Appendix A. (**B**) Boxplot of Sequential Organ Failure Assessment (SOFA) score and Acute Physiology and Chronic Health Evaluation II (APACHE) among patients with different vaccination statuses and ages. (**C**) Heatmap of the O_2_ requirement among patients with different vaccination statuses. HFNC: high-flow nasal cannula.

**Figure 2 vaccines-10-01181-f002:**
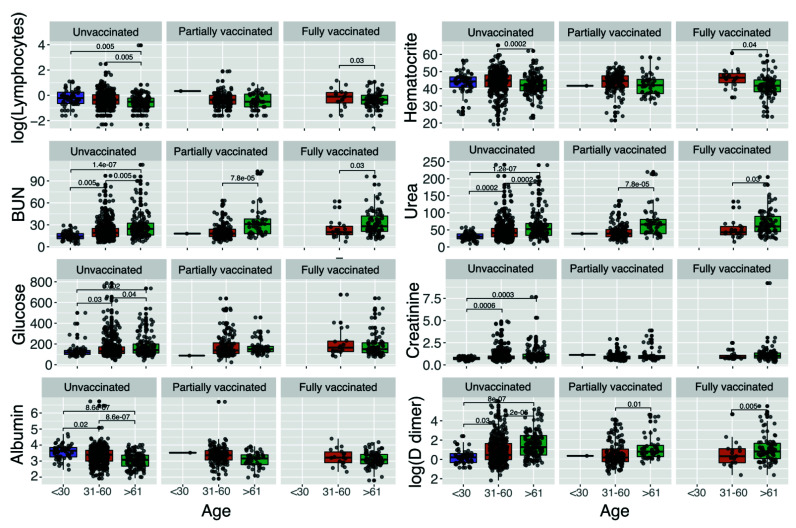
Laboratory parameters that significantly vary among vaccination status and age group. We used logarithmic distribution in the lymphocytes and D-dimer parameters for visualization purposes. Statistically significant differences are given by Wilcoxon rank-sum test. Units: lymphocytes (10^3^/mm^3^), blood urea nitrogen (BUN) (mg/dL), glucose (mg/dL), albumin (g/dL), hematocrit (%), urea (mg/dL), creatinine (mg/dL), D-dimer (mg/dL).

**Figure 3 vaccines-10-01181-f003:**
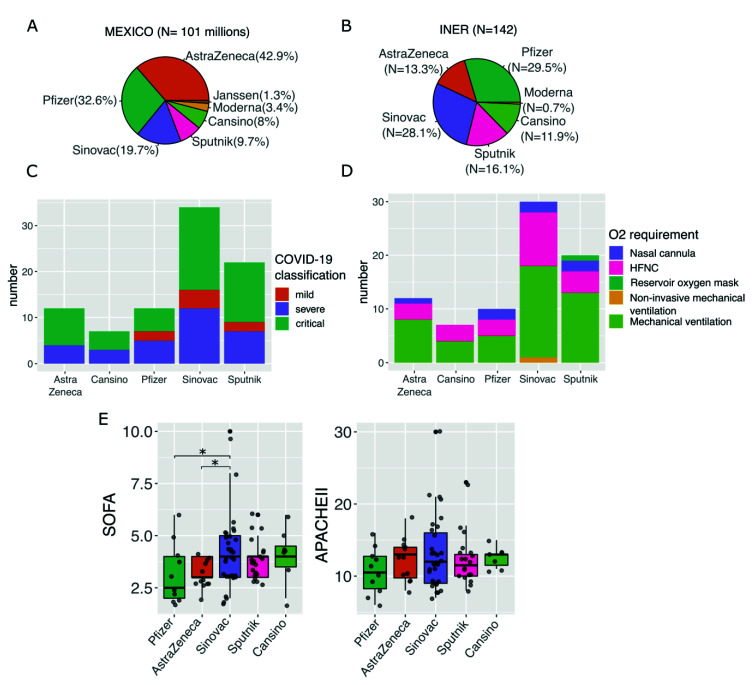
Distribution of vaccine strategies and their impact on disease severity in fully vaccinated patients. (**A**) Pie chart of the proportion of the vaccines applied at the national level. (**B**) Pie chart of the proportion of the different vaccines applied at the INER. (**C**) Bar plot shows the proportion of COVID-19 classification in patients with different vaccines. (**D**) Bar plot depicting the proportion of O_2_ requirement in patients with different vaccines. (**E**) Boxplot of Sequential Organ Failure Assessment (SOFA) score and Acute Physiology and Chronic Health Evaluation II (APACHE) among patients with different vaccines. Statistically significant differences are available in Appendix A. In (**C**–**E**) panels, only patients > 61 years old were included. * *p* < 0.05.

**Figure 4 vaccines-10-01181-f004:**
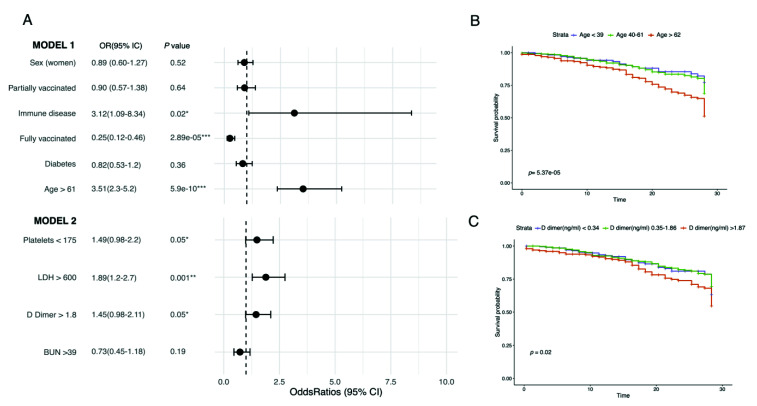
Variables affecting outcome in COVID-19 patients. (**A**) Forest plot of the two fitted generalized linear model (GLM) representing the odds ratio and 95% CI for each variable. Estimates are available in Appendix A. (**B**) Kaplan–Meier curve depicting the survival probability of patients with different age ranges. (**C**) Kaplan–Meier curve depicting patients’ survival probability with different D-dimer values. Kaplan–Meier curves were constructed using days of hospitalization as a time variable. Both models were constructed with the outcome (deceased or recovered) standardized at 28 days. Units: platelets (×10^9^/L), lactate dehydrogenase (LDH) (µg/L), D-dimer (µg/dL), and blood urea nitrogen (BUN) (mg/dL). * *p* ≤ 0.05, ** *p* ≤ 0.001, *** *p* ≤ 0.0001.

**Figure 5 vaccines-10-01181-f005:**
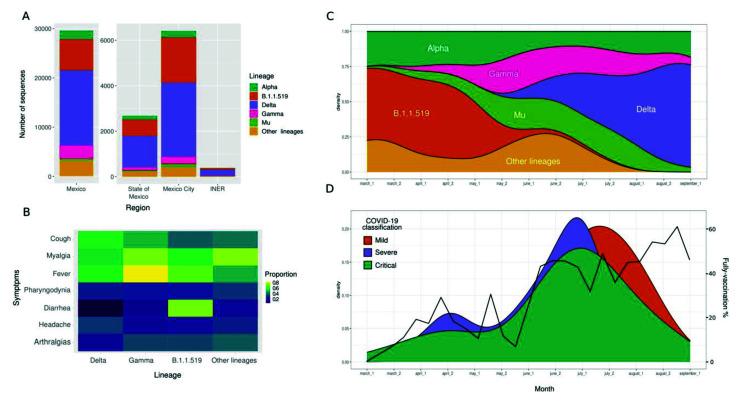
SARS-CoV-2 lineage distribution and its impact on disease severity. (**A**) Distribution of SARS-CoV-2 lineages at the country level (Mexico), metropolitan area (the State of Mexico and Mexico City), and our research center (INER). (**B**) Heatmap of the principal symptoms among patients infected with the different SARS-CoV-2 lineages. (**C**) Dominance of lineages in our cohort over time (months). (**D**) Distribution of patients with mild, severe, or critical COVID-19 classification and percentage of fully vaccinated patients in our cohort over time (months).

**Table 1 vaccines-10-01181-t001:** Demographics of the cohort.

Vaccination Status	Outpatients (*n* = 111)				Hospitalized (*n* = 855)			
	Unvaccinated (*n* = 39) [35.1%]	Partially Vaccinated (*n* = 32) [28.8%]	Fully Vaccinated (*n* = 40) [36%]	*p*	Unvaccinated (*n* = 572) [66.9%]	Partially Vaccinated (*n* = 178) [20.8%]	Fully Vaccinated (*n* = 105) [12.2%]	*p*
**Age**								
<30, *n* (%)	13 (33.3%)	4 (12.5%)	14 (35%)	0.03	54 (9.4%)	1 (0.5%)	0	<0.001
31–60, *n* (%)	25 (64.1%)	22 (68.7%)	18 (45%)	0.04	382 (66.7%)	132 (74.1%)	21 (20%)	<0.001
=/> 60, *n* (%)	1 (2.50%)	6 (18.7%)	8 (20%)	0.03	135 (23.6%)	45 (25.2%)	82 (78%)	<0.001
NA	0	0	0		1	0	2	
**Gender**								
Female, *n* (%)	17 (43.5%)	20 (62.5%)	16 (40%)	ns	200 (34.9%)	65 (36.5%)	52 (49.5%)	0.02
NA	0	0	0		0	0	0	
**Vaccine type**								
Total of vaccinated patients		32 (100%)	40 (100%)			179 (100%)	104 (100%)	
Pfizer, *n* (%)	na	12 (37%)	28 (70%)	0.01	na	34 (19%)	14 (13.3%)	ns
AztraZeneca, *n* (%)	na	13(40.6%)	4 (10%)	0.02	na	69 (38.7%)	15 (14.2%)	<0.001
SinoVac, *n* (%)	na	3 (9.3%)	5 (12%)	ns	na	18 (10.1%)	35 (33.3%)	<0.001
Sputnik, *n* (%)	na	2 (6.2%)	2 (5%)	ns	na	51 (28.6%)	21 (20%)	ns
Cansino, *n* (%)	na	0	1 (2.5%)	na	na	na	18 (17.1%)	na
J&J, *n* (%)	na	0	0	na	na	1 (0.56%)	0	ns
Moderna, *n* (%)	na	0	0	na	na	0	1 (0.95%)	ns
NA	na	2	0		na	5	1	
**Comorbidities**								
Diabetes, *n* (%)	2 (5.1%)	4 (12.5%)	1 (2.5%)	ns	115 (20.1%)	44 (24.7%)	42 (40%)	<0.001
Hypertension, *n* (%)	5 (12.8%)	4 (12.5%)	5 (12.5%)	ns	147 (25.6%)	45 (25.2%)	53 (53.3%)	<0.001
Obesity, *n* (%)	5 (12.8%)	7 (21.8%)	4 (10%)	ns	255 (44.5%)	82 (46%)	47 (44.7%)	ns
Smoking, *n* (%)	2 (5.1%)	1 (3.1%)	2 (5%)	ns	177 (30.9%)	57 (32%)	25 (23.8%)	ns
COPD, *n* (%)	0	0	2 (5%)	na	12 (2%)	2 (1.1%)	4 (3.8%)	ns
Immune disease, *n* (%)	0	0	2 (5%)	na	12 (2%)	1 (0.5%)	3 (2.8%)	ns
NA	89	85	105		33	13	10	
**Number of comorbidities**								
None, *n* (%)	22 (56.4%)	13 (40.6%)	25 (62.1%)	0.05	97 (16.9%)	34 (19.1%)	10 (9.5%)	0.05
1, *n* (%)	9 (23.07%)	2 (6.2%)	4 (10%)	ns	217 (37.9%)	58 (32.5%)	21 (20%)	<0.001
2, *n* (%)	2 (5.1%)	1 (3.1%)	4 (10%)	ns	170 (29.7%)	55 (30.8%)	35 (33.3%)	ns
=/> 3, *n* (%)	1 (2.5%)	3 (9.3%)	2 (5%)	ns	87 (15.2%)	31 (17.4%)	39 (37.1%)	<0.001
NA	5	13	5		1	0	0	
**In-hospital treatment**								
Dexamethasone, *n* (%)	5 (12.8%)	2 (6.2%)	1 (2.5%)	ns	554 (96.8%)	169 (94.9%)	98 (93.3%)	ns
Remdesivir/Baricitinib, *n* (%)	0	0	0	na	11 (1.9%)	3 (1.6%)	3 (2.8%)	0.02
Remdesivir/Dexamethasone, *n* (%)	0	0	0	na	4 (0.6%)	0	1 (0.95%)	ns
Dexamethasone/Baricitinib, *n* (%)	0	0	0	na	1 (0.1%)	2 (1.1%)	0	ns
NA	5	13	6		0	0	2	
**Previous steroid**								
Dexamethasone, *n* (%)	0	0	0	na	202 (35.3%)	66 (37%)	40 (38%)	ns
Prednisone, *n* (%)	0	0	0	na	28 (4.8%)	0	5 (4.7%)	0.001
Betamethasone, n (%)	0	0	0	na	12 (2%)	7 (3.9%)	2 (1.9%)	ns
Dexamethasone/Prednisone, *n* (%)	0	0	0	na	6 (1.04%)	2 (1.1%)	0	ns
Dexamethasone/Betamethasone, *n* (%)	0	0	0	na	5 (0.87%)	4 (2.2%)	0	ns
NA	5	13	6		0	0	2	
**Symptoms**								
Fever, *n* (%)	19 (48.7%)	10 (31.2%)	12 (30%)	0.01	409 (71.5%)	134 (75.2%)	57 (54.2%)	<0.001
Cough, *n* (%)	19 (48.7%)	19 (59.3%)	15 (37.5%)	ns	333 (58.2%)	107 (60.1%)	68 (64.7%)	ns
Diarrhea, *n* (%)	4 (10.2%)	1 (3.1%)	4 (10%)	0.01	61 (10.6%)	17 (9.5%)	6 (5.7%)	ns
Myalgias, *n* (%)	9 (23%)	7 (21.8%)	11 (27.5%)	ns	382 (66.7%)	111 (62.3%)	65 (61.9%)	ns
Arthralgias, *n* (%)	10 (25.6%)	10 (31.2%)	8 (20%)	0.002	163 (28.4%)	47 (26.4%)	24 (22.8%)	ns
Nasal congestion, *n* (%)	0	0	2 (5%)	na	51 (8.9%)	21 (11.7%)	11 (10.4%)	ns
Pharyngodynia, *n* (%)	7 (17.9%)	7 (21.8%)	11 (27.5%)	ns	140 (24.4%)	57 (32%)	29 (27.6%)	0.05
Anosmia, *n* (%)	2 (5.1%)	3 (9.3%)	2 (5%)	ns	43 (7.5%)	14 (7.8%)	3 (2.8%)	ns
Headache, *n* (%)	13 (33.3%)	7 (21.8%)	13 (32.5%)	<0.001	190 (33.2)	60 (33.7%)	33 (31.4%)	ns
NA	5	20	14		0	0	18	

COPD: chronic obstructive pulmonary disease. NA: data not available. *p* values were obtained from Chi-square or Fisher’s exact test. ns = non-significant, na = no available comparison.

**Table 2 vaccines-10-01181-t002:** Outcome, severity indexes, and O_2_ requirement among hospitalized patients with different vaccination statuses.

Vaccination Status	Hospitalized (*n* = 855)			
	Unvaccinated (*n* = 572) [66.9%]	Partially Vaccinated (*n* = 178) [20.8%]	Fully Vaccinated (*n* = 105) [12.2%]	*p* Value
**Outcome**				
Recovered, n (%)	433 (71.6%)	139 (78%)	88 (83.8%)	0.05
Deceased, n (%)	129 (22.5%)	37 (20.7%)	16 (15.2%)	ns
NA	10	2	1	
**Severity indexes**				
APACHE II, med(IQR)	8.5 (6–11)	9 (6.2–11)	11 (9–14)	<0.001
SOFA, med(IQR)	3 (3–4)	3 (3–4)	4 (3–4)	ns
GLASGOW, med(IQR)	15 (15–15)	15 (15–15)	15 (15–15)	ns
**O_2_ requirement**				
Nasal cannula, *n* (%)	54 (9.4%)	19 (10.6%)	12 (11.4%)	ns
HFNC, *n* (%)	135 (23.6%)	50 (28%)	28 (26.6%)	ns
Mechanical ventilation, *n* (%)	375 (65.5%)	105 (58.9%)	61 (58%)	ns
Reservoir oxygen mask, *n* (%)	7 (1.2%)	3 (1.6%)	1 (0.9%)	ns
Non-invasive mechanical ventilation, *n* (%)	1 (0.1%)	1 (0.5%)	1 (0.9%)	ns
NA	0	0	2	

NA: data not available. *p* values were obtained from Chi-square or Fisher’s exact test. ns = non-significant, na = no available comparison.

## Data Availability

The genomic information generated during the current study is available in the GenBank repository, accession numbers: ON158371–ON158443. The clinical datasets used during the current study are available from the corresponding author on reasonable request.

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
