# Peer review of "Clinical and Virological Features of Patients Hospitalized with Different Types of COVID-19 Vaccination in Mexico City"

_vaccines, 2022, doi:10.3390/vaccines10081181_

Round 1

Reviewer 1 Report

This study explores clinical and virological features of patients hospitalized with different types of COVID-19 vaccination in Mexico City. The paper sounds interesting, quite organized and comprehensive. Vaccination against COVID-19 has raised many concerns in public opinion. I think that it is a very relevant topic that must be addressed. The design of the study is good. I only have some suggestions:

- This study suggests that vaccination is essential to reduce mortality in a comorbid population such as that of Mexico. I strongly agree with the author, but some special populations should be discussed (lines 57-59). Indeed, ref 8 is not enough robust to support this statement. What about patients affected by autoimmune disease? Some considerations need to be discussed (i.e. Myasthenia gravis, thyroid disorders, etc). Consider the role of vaccination in special populations and autoimmune neuromuscular disease such as myasthenia gravis. Read and cite recent papers on this topic.

- although most of the patients were subjected to mechanical ventilation, the highest proportion of intubated patients was in the unvaccinated group. Conversely, fully vaccinated was associated with improved outcomes (line 223) and number of critical and severe patients seem to decrease as the vaccination increases (line 246). However, I do not think that hypertension and diabetes might have a role on this. Among unvaccinated patients, some myasthenic patients might account for a relevant percentage of patients needing ventilation. Recent evidence shows a very different outcome in MG patients depending on vaccination (see previous discussed paper, Lupica et al 2022). Other comorbidities to include are respiratory comorbidity and guillan-barré syndrome.

-table 1 shows that the highest proportion of critical patients was unvaccinated older than 61 years old. This underlines the role of comorbidities in COVID-19 outcome. 

-the design of the study is in line with the aims. The results have been discussed in full.

-there are no relavant grammar mistakes. 

Author Response

REVIEWER 1

This study explores clinical and virological features of patients hospitalized with different types of COVID-19 vaccination in Mexico City. The paper sounds interesting, quite organized and comprehensive. Vaccination against COVID-19 has raised many concerns in public opinion. I think that it is a very relevant topic that must be addressed. The design of the study is good. I only have some suggestions:

- This study suggests that vaccination is essential to reduce mortality in a comorbid population such as that of Mexico. I strongly agree with the author, but some special populations should be discussed (lines 57-59). Indeed, ref 8 is not enough robust to support this statement. What about patients affected by autoimmune disease? Some considerations need to be discussed (i.e. Myasthenia gravis, thyroid disorders, etc). Consider the role of vaccination in special populations and autoimmune neuromuscular disease such as myasthenia gravis. Read and cite recent papers on this topic.

R= We agree with the reviewer comment and we added a paragraphs both, in introduction (Lines 81-91) and discussion (Lines 370-372) sections about autoimmune diseases and vaccination.

- although most of the patients were subjected to mechanical ventilation, the highest proportion of intubated patients was in the unvaccinated group. Conversely, fully vaccinated was associated with improved outcomes (line 223) and number of critical and severe patients seem to decrease as the vaccination increases (line 246). However, I do not think that hypertension and diabetes might have a role on this. Among unvaccinated patients, some myasthenic patients might account for a relevant percentage of patients needing ventilation. Recent evidence shows a very different outcome in MG patients depending on vaccination (see previous discussed paper, Lupica et al 2022). Other comorbidities to include are respiratory comorbidity and guillan-barré syndrome.

R= In the same way, we added a paragraph in discussion section arguing the importance of hypertension and diabetes in severe and fatal outcomes (Lines 364-367). Also we mentioned the findings of Lupica et.al. about the different outcome in MG patients depending on vaccination (Lines 369-372).

-table 1 shows that the highest proportion of critical patients was unvaccinated older than 61 years old. This underlines the role of comorbidities in COVID-19 outcome. 

R= Among all of the risk factors that have been described for severe COVID-19, age above 60 years old has been the most relevant factor, followed by the presence of comorbidities. Comorbidities can affect the outcome of these patients, as risk factors accumulate, prognosis worsens for this patients. Table 1 shows that most of the vaccinated patients (78%) were above 60 years old and this group of patients had comorbidities, with 37% of this group with 3 or more concomitant conditions.  These findings highlight the protective effect of vaccination in this population, even with advanced age and comorbidities, vaccination avoided critical COVID-19 disease of even fatal outcome.

Reviewer 2 Report

The manuscript entitled " Clinical and Virological features of patients hospitalized with 2 different types of COVID-19 vaccination in Mexico City". Title, abstract and overall rationale of work is satisfactory and very well presentation and design. However, there are still some minor concerns, which needs to be addressed before publication.

1) Keywords: Author should be revised keywords and add suitable keywords at least five.

2) Introduction section is too short and author need to revise and make a more details about this related study.

3) Author must be enhance the quality of the all figures

4) I would suggest the authors to enhance your theoretical discussion and arrives your debate or argument.

5) Author must be incorporate conclusion section in the revise manuscript.

Author Response

REVIEWER 2

The manuscript entitled " Clinical and Virological features of patients hospitalized with 2 different types of COVID-19 vaccination in Mexico City". Title, abstract and overall rationale of work is satisfactory and very well presentation and design. However, there are still some minor concerns, which needs to be addressed before publication.

1) Keywords: Author should be revised keywords and add suitable keywords at least five.

R= Thank you for your comment. We have revised the keywords and added one.

2) Introduction section is too short and author need to revise and make a more details about this related study.

R= Thank you for your suggestion. We included relevant information in the introduction section. In particular, we added background regarding SARS-CoV-2 variants and it´s WHO classification (Lines: 58-66), as well as some information about a particular lineage that was predominant in Mexico during our study (Lines: 72-75). Also, we included a paragraph mentioning the importance of comorbidities and vaccination, which we believe is necessary to support our aims (Lines: 81-91).

3) Author must be enhance the quality of the all figures

R= We thank the reviewer for noticing this mistake. In the revised version of the manuscript we have revised and improved the resolution of all figures.

4) I would suggest the authors to enhance your theoretical discussion and arrives your debate or argument.

R= Thank you for your suggestion. In the revised version of the manuscript we have discussed thoroughly all of our findings. Particularly, we elaborate in the discussion about the role of comorbidities in COVID-19 severity and the particular context of the Mexican population (Lines:364-372 ). Also, we have included examples of the negative effect of advanced ages in other infectious diseases (Lines:373-380 ). Moreover, we included the potential implications of the high levels of markers of disease severity for COVID-19 patients (Lines: 384-388 ). Also, we suggested potential explanations of the tendency of lower severity that we found in patients vaccinated with Pfizer (Lines: 398-402 ). Finally, we included a paragraph about differences in symptomatology associated with SARS-CoV-2 lineages (Lines: 415-425) and broad our discussion on the risk factors for severe disease in our cohort (Lines: 427-432). 

5) Author must be incorporate conclusion section in the revise manuscript.

R=  Thank you for your suggestions. In the revised version of the manuscript we have clearly stated the conclusions and included a sentence on SARS-CoV-2 lineages (Lines:440-453 )

Round 2

Reviewer 2 Report

Author fulfill all comments raised by the previous version. I recommend for publication.

This manuscript is a resubmission of an earlier submission. The following is a list of the peer review reports and author responses from that submission.